# Zeaxanthin dipalmitate-enriched wolfberry extract improves vision in a mouse model of photoreceptor degeneration

Xiongmin Chen[1☯], Sensen Zhang[1☯], Lili Yang[2☯], Qihang Kong[3], Wenhua Zhang[2], Jinhong Zhang[2], Xiangfeng Hao[2], Kwok-Fai So[1,4,5]*, Ying Xu[1,4]*

1 Key Laboratory of CNS Regeneration (Jinan University)-Ministry of Education, Guangdong-Hong Kong-Macau Institute of CNS Regeneration, Guangdong Key Laboratory of Non-Human Primate Research, Jinan University, Guangzhou, Guangdong Province, China, 2 Bairuiyuan Gouqi Corp., Yinchuan, Ningxia Province, China, 3 Department of Ophthalmology, The First Affiliated Hospital of Jinan University, Guangzhou, Guangdong Province, China, 4 Co-Innovation Center of Neuroregeneration, Nantong University, Nantong, Jiangsu Province, China, 5 State Key Laboratory of Brain and Cognitive Sciences, Hong Kong Special Administrative Region, Hong Kong, China

☯ These authors contributed equally to this work.
* hrmaskf@hku.hk (K-FS); xuying@jnu.edu.cn (YX)

**Data Availability Statement:** All relevant data are within the article and its Supporting Information files.

## Abstract

Zeaxanthin dipalmitate (ZD) is a chemical extracted from wolfberry that protects degenerated photoreceptors in mouse retina. However, the pure ZD is expensive and hard to produce. In this study, we developed a method to enrich ZD from wolfberry on a production line and examined whether it may also protect the degenerated mouse retina. The ZD-enriched wolfberry extract (ZDE) was extracted from wolfberry by organic solvent method, and the concentration of ZD was identified by HPLC. The adult C57BL/6 mice were treated with ZDE or solvent by daily gavage for 2 weeks, at the end of the first week the animals were intraperitoneally injected with N-methyl-N-nitrosourea to induce photoreceptor degeneration. Then optomotor, electroretinogram, and immunostaining were used to test the visual behavior, retinal light responses, and structure. The final ZDE product contained ~30mg/g ZD, which was over 9 times higher than that from the dry fruit of wolfberry. Feeding degenerated mice with ZDE significantly improved the survival of photoreceptors, enhanced the retinal light responses and the visual acuity. Therefore, our ZDE product successfully alleviated retinal morphological and functional degeneration in mouse retina, which may provide a basis for further animal studies for possible applying ZDE as a supplement to treat degenerated photoreceptor in the clinic.

## Introduction

Photoreceptor degenerative diseases are a group of retinal diseases including retinitis pigmentosa (RP), age-related macular degeneration (AMD), and other inherited retina dystrophies. In those diseases, photoreceptors degenerate and lose the ability to transfer light into electrical signals thus leading to blindness finally [1]. Among those diseases, RP is one of the most

**Funding:** Ningxia Key Research and Development Program Grant (No. 2021BEF02040 to WZ), the Natural Science Foundation of Guangdong Province (2023A1515012397 to YX). The funders had no role in study design, data collection and analysis, decision to publish, or preparation of the manuscript.

**Competing interests:** WZ, LY, JZ, and XH are the employees of Bairuiyan Gouqi Corp and are involved in the patent (Application No. 201911029950.6, name as "A method to produce zeaxanthin dipalmitate") listed in the text. The other authors have no conflict of interest to declare. This does not alter our adherence to PLOS ONE policies on sharing data and materials.

common forms of inherited retinal disease that causes degeneration and death of cone and rod cells, affecting approximately 1 in 4,000 individuals [2]. Many strategies have been applied to preserve or replace photoreceptors, including antioxidant or anti-inflammatory agents, gene therapy, stem cell therapy, retinal prosthesis therapy, etc. [3]. Among them, some have entered clinical trials with promising results and FDA approved products like Luxturna as a gene therapy for Leber's congenital amaurosis type 2 [4], and electric transplant Argus II retinal prosthesis [5]. While these strategies may offer effective long-term treatment, the expensive cost and risk of surgery limits the application, therefore new treatment option for degenerative retina is still in great need.

Many researchers have explored the protective effect on the degenerated retina from plant extract, like curcumin, flavonoids including luteolin, *Ginkgo Biloba* extract and green tea extract, resveratrol, forskolin, saffron, and Lycium barbarum etc [6]. Among them, lycium barbarum or wolfberry is a traditional Chinese herb that nourishes the kidney, liver, and eyes [7]. Its extract has demonstrated a protective effect in retinal diseases, including animal models of glaucoma [8], retinitis pigmentosa [9,10], and diabetic retinopathy [11], as well as RP patients [12]. Zeaxanthin dipalmitate (ZD) is a major carotenoid in wolfberry extract (structure shown in Fig 1A), which has a strong antioxidant function. Studies have shown that ZD has a strong ability to scavenge free radicals and protect liver cells against several liver diseases [13,14]. Our previous study has proved that ZD can delay photoreceptor degeneration in rd10, a genetic mutation mouse model of RP by just one dose of intravitreous injection [15]. And ZD reduced the expression of genes that are involved in inflammation, apoptosis and oxidative stress and inhibited the STAT3, CCL2 and MAPK pathways. Therefore ZD may serve as a potential candidate to treat RP. But intravitreous injection causes damage to the eye and can't be applied repetitively, oral feeding of ZD is preferred in the clinic. On the other hand, oral taking of ZD requires a large amount of ZD, purification which is time-consuming and expensive. Therefore, ZD-enriched agent may have advantages over pure ZD in the oral application due to the ability of mass production therefore lower price. In this study, we developed a method to

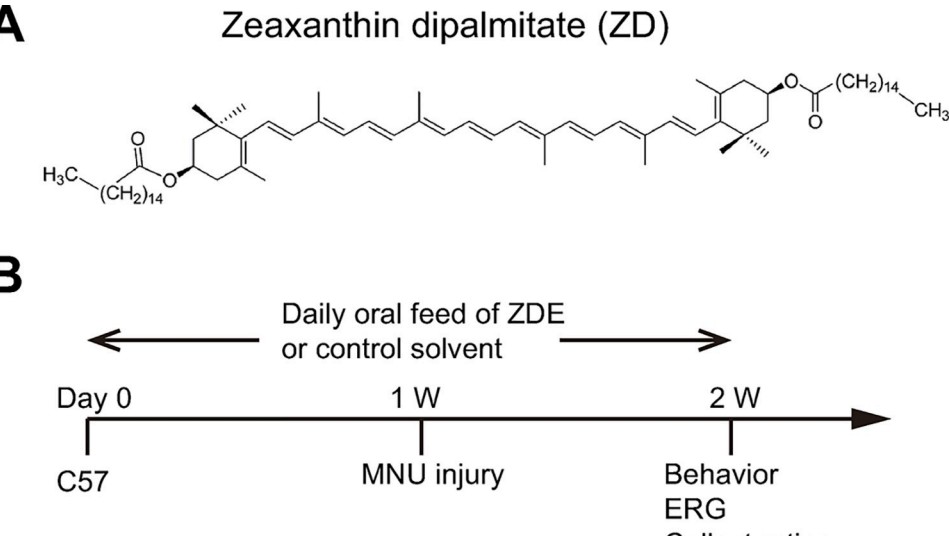

**Fig 1. Experiment protocol. (A)**. Chemical structure of ZD. **(B)**. Adult C57 mice were fed with ZD-enriched wolfberry extract (ZDE) orally daily for two weeks, during which 40 mg/kg MNU was intraperitoneally injected at 1 week to induce retinal photoreceptor cell degeneration. Then the visual behavior and ERG were tested at 2 weeks after treatment before the animals were sacrificed and the retina collected for immunostaining.

enrich ZD from wolfberry, which enabled us to administrate the ZD agent by daily oral administration.

We further explored the protective effect of ZD-enriched wolfberry extract (ZDE) on the degenerated retina of mice induced by N-methyl-N-nitrosourea (MNU). MNU is an alkylated toxic substance that induces specific apoptosis of photoreceptors within a week after injection. Unlike rd10 that the photoreceptors degenerate since postnatal 17 (P17) and most rods die by P25, which request the strict timing of treatment, MNU can induce a fast degeneration of photoreceptors in adult mice at any time after injection, therefore MNU-injured mouse model widely serves as a chemical-induced photoreceptor degenerative model [16].

## Materials and methods

### Animals

Male C57BL/6J (C57) mice were purchased from Liaoning Changsheng Biotechnology Co., LTD. All mice were maintained under standard laboratory conditions in the animal facility at Jinan University (room temperature range between 18˚C-23˚C, humidity between 40–65%, dark/light cycle of 12/12 hours), and the animals had free access to regular food and water. All animal experiments were conducted following the ARVO Statements for the Use of Animals in Ophthalmic and Vision Research and the animal study protocol was approved by the Laboratory Animal Ethics Committee of Jinan University (#IACUC-20210706-03 on July 6th, 2021). All efforts were made to minimize the number of animals used and their suffering, including operated by skillful researchers, careful design of the experiment and special animal care.

### Extraction of ZDE from wolfberry

The wolfberry fruit (*Lycium barbarum L.*) was the No. 7 product collected from the farm located at N36˚45'-39˚30', E 105˚16'-106˚80' at Zhongning, Ningxia, China, with batch number WG20072815TY270C, and passed the standard inspection GB/18672 Gouqi. A certain amount of the dried wolfberry was soaked in water with 5 times its volume (W/V) for 2 hours, then crushed with a Cyclone universal pulverizer for 30 seconds, and the same volume of water was added again at room temperature. The mixture was ultrasonicated for 60 minutes, and centrifuged at 5000g for 10 minutes to obtain the polysaccharide extract (in the supernatant) and the precipitate.

The precipitate was collected and dried at 40˚C and then crushed again to obtain the coarse residue of wolfberry. ZDE was extracted from the coarse residue of wolfberry by organic solvent method with a self-developed optimal process condition. Briefly, the coarse wolfberry residue was mixed with a mixed solvent (hexane: ethanol = 2.6:1, V: V, patent formula, Patent application No. CN 201911029950.6) with a solid-to-liquid ratio of 1:20 (W/V), and extracted at 50˚C for 1 hour. The extract was filtered through a 150-mesh sieve, the filtrate was collected, and the filter residue was extracted once more with the above protocol. The two filtrates were combined and concentrated by spinning at 60˚C with a negative pressure of 0.08 Mpa till no solvent came out. Finally, an oily orange solid was obtained, which was the product of ZDE.

### Measurement of ZD concentration by HPLC

The concentration of ZD in the wolfberry extracts was measured by High-Performance Liquid Chromatography (HPLC) according to a group standard (T/NXFSA 004S—2020). Briefly, the ZD standard (with purity >95%) was dissolved in 1:1 (v/v) mobile phase A solution (methanol:

acetonitrile: water, 81:14:5): mobile phase B solution (dichloromethane) to make the standard solution.

To prepare the test sample from the dry fruit of wolfberry, 10g of frozen crushed wolfberry was soaked in water at 5 times its volume (W/V) for 2 hours, then crushed with a pulverizer and centrifuged at 3000r/min for 5 minutes. The precipitate was collected, dried, and crushed. Then 0.2g crushed sample was weighed (accurate to 0.0001g) and put in a mortar, added Quartz sand (1/4 of the sample weight), and grinding extracted with hexane: ethyl acetate: methanol (1:1:1, v/v/v) multiple times, with 1-3ml solution collected each time till the solution became transparent. All the extracted solution was then combined and fit to the volume of 50ml with 100% ethanol. After centrifuging the solution at 1000 r/min for 3 minutes, the precipitate was discarded and the supernatant was collected. The supernatant from the ground dry fruit of wolfberry then was solved in n-hexane: ethyl acetate: methanol (1:1:1, v/v/v) and collected for the HPLC test.

To prepare the test sample from the ZDE product, 0.2g of the sample (accurate to 0.0001g) was dissolved in n-hexane: 100% ethanol (3:1, v/v) to a fixed volume of 50ml. The solution was centrifuged at 1000r/min for 3 min and the supernatant was collected for the HPLC test.

To run the HPLC test, a 10ul sample was added into a C30 column (4.6mm inner diameter, 250mm column length, and the particle size of 5 mm) and the Chromatographic separation was performed at 30˚C. The mobile phase consisted of mobile phase A (methanol: acetonitrile: water, 81:14:5) and B solution (dichloromethane). The mobile phase composition started at 84% mobile phase A for 3 min, then 83% for 20 mins, then 45% for 15 mins, 25% for 18 mins, and back to 84% for A solution. The flow rate was 1.0 ml/min and the detection wavelength was set at 450 nm.

## Experimental design and drug application

The final product of ZDE was dissolved in corn oil at the following: to make a 1mg/kg body weight ZDE solution, 0.1g extract (which contained 3mg ZD) was dissolved in 30 ml pure corn oil, and mouse was oral feed with the solution at the volume of 10ml/kg.

For the treatment, animals were randomly assigned to two groups: MNU-injured mice treated with ZDE or solvent. Animals were pretreated with ZDE or solvent for a week by daily oral feeding, then MNU solution (40mg/kg in normal saline solution) was injected intraperitoneally to induce specific photoreceptor degeneration [17–19]. Then animals were fed with ZDE for another week before visual behavior and ERG were tested. Then animals were sacrificed by cervical dislocation after ERG recording and retinas were collected. Normal mice without any treatment were included as control group. Normal mice treated with the solvent or ZDE were also tested to exclude the possible side effect on the normal retina. The detailed protocol is illustrated in Fig 1B.

In a preliminary experiment, we screened for the safe dose of ZDE at 1, 3, 9, 27, 54, 100, and 200mg/kg body weight by daily oral feeding the animal for 2 weeks, and found that the body weight of animals remained stable for all doses (S1 Fig). Then we tested the protective doses of ZDE at 1, 3, 9, and 27 mg/kg body weight by examining the thickness of ONL where photoreceptors are located. We identified the best protective effect of ZDE at 9 mg/kg. Then for the following experiments, we applied ZDE at 9mg/kg to examine its effect on the visual behavior and retinal light responses.

## Visual behavioral tests

The visual behavior of mice was examined by both the dark-light transition test and the opto-motor systems at the end of the treatment. The dark-light transition test measures the

tendency of a mouse to stay in darkness rather than in an illuminated area and it was conducted as previously described [20]. Briefly, the light and dark chambers were connected through an open door; a mouse was placed in the center of the illuminated white chamber and could move freely between the chambers. The movement was recorded by cameras installed in both chambers and connected to a recorder (Noldus, Wageningen, the Netherlands). The duration a mouse spent in the dark chamber during a 5-minute test was quantified automatically by the EthoVision XT 8.0 software (Noldus).

The optomotor test measures visual acuity by observing the head turning in response to moving gratings as we described before [21]. Briefly, the mice were placed freely on a high-center platform surrounded by computer screens that displayed vertical rotating sine gratings. The gratings were programmed by Matlab (MATLAB 8.0, MathWorks, Natick, MA, USA) and had a 100% contrast and moving speed at 12 cycles/s, with increasing spatial frequencies ranging from 0.1 to 0.5 cycles/degree. Animals reflexively track the gratings by moving their head (i.e. optokinetic reflex) as long as they can see them. Head movements were videotaped and the maximal spatial frequency at which an optokinetic response could be observed was manually recorded to reflect the visual acuity of a mouse.

### Electroretinogram (ERG)

To evaluate the retinal light responses, the mice were dark-adapted overnight and ERG experiments were performed by RETI-scan system (Roland Consult, Brandenburg, Germany) as we previously described [9]. Briefly, mice were anesthetized with tribromoethanol (0.2 ml/10g of 1.25% solution) and placed on a heated platform (37°C) under dim red light. Pupils were dilated with phenylephrine-HCl (0.5%) and tropicamide (0.5%). ERGs were recorded with gold-plated wire loop electrodes contacting the corneal surface as the active electrode. Stainless steel needle electrodes were inserted into the skin near the eye and the tail serving as the reference and ground leads, respectively. Dark-adapted mice were first stimulated by green flashes of 0.01, 0.1, and 3.0 cd.s/m$^2$ to record the scotopic responses. Then the mice were light-adapted for 5 mins using a green background (20 cd/m$^2$), and photopic responses to the green flash of 3.0 and 10.0 cds/m$^2$ were recorded. ERG data were collected using the amplifier of the RETI-scan system at a sampling rate of 2 kHz, and subsequently analyzed with RETIport software (Roland Consult) after applying 50-Hz low-pass filtering. The a-wave amplitude was determined from the baseline to the first negative peak, and the b-wave amplitude was measured from the a-wave trough to the subsequent positive peak. For each mouse, the responses of the two eyes were averaged as a data point.

### Tissue processing

After ERG testing, the mice were sacrificed using an overdose of anesthetic (intraperitoneal injection of 100 mg/kg pentobarbital sodium; R&D Systems, Minneapolis, MN, USA). Both eyes were removed and placed in 4% paraformaldehyde (PFA) at room temperature for 30 minutes. Tissues were washed 3 times with PBS buffer solution for 5 mins each time and cryo-protected overnight in PBS containing 30% sucrose solution. The tissues were then embedded in optimal cutting temperature compounds (Tissue Tek, Torrance, CA, USA) and cryosectioned on a microtome (Leica Microsystems, Wetzlar, Germany) through the optic disk longitudinally at a thickness of 14 μm. The retinal sections were then incubated at room temperature for 5 min with 4',6-diamidino-2-phenylindole (DAPI,1:1000, Electron Microscopy Sciences, Hatfield, PA, USA) and then washed, mounted on glass slides, and sealed.

## Image collection and processing

DAPI-stained tissues were imaged with a fluorescence microscope (Carl Zeiss). To assess the photoreceptor survival, the thickness of the outer nuclear layer (ONL) where photoreceptor soma is located was measured by Image J software, and the number of nuclei layers in the ONL row was counted. Due to the uneven degeneration of photoreceptors from the central to peripheral [16], we measured the thickness and the number of cell layers of ONL at 400, 800, 1200 and 1600 μm away from the central point of the optic nerve on both sides and then averaged to get the values for each locations of the section. For the dose-dependent curve, the ONL thickness measured around 1000 μm was measured and compared among all doses. For each retina, the thickness and number of layers of ONL from 3–5 cryo-sections collected at different locations in the eye cup was averaged to obtain a data point for this animal, and these values were then averaged to obtain a mean value for the group.

## Statistical analysis

All data were expressed as means ±SEMs and analyzed by Prism 7 (GraphPad Software, San Diego, CA, USA). One-way or two-way analysis of variance (ANOVA) followed by post hoc tests were performed. P value<0.05 indicated a significant difference. N represents the total number of animals examined in each group.

## Results

### ZDE contained 9-fold higher ZD than raw fruit

By HPLC test, we compared the concentration of pure ZD in dry raw fruit of wolfberry and our ZED product. The ZD component was identified by the peak arose around 28 mins, which was similar to the ZD standard (Fig 2A). The spectrum showed that the area of ZD peak was 9.34 times higher in ZED product than in wolfberry (Fig 2). Fitting the area to the standard curve set up with pure ZD showed that the concentration of ZD was 3.12 ± 0.06 mg/g in wolfberry and 28.61 ± 0.64 mg/g in ZDE product from 4 repetitive measurements. Therefore, the method developed by us enriched ZD with 9-fold higher concentration than the raw fruit.

### ZDE improves the survival of photoreceptors in MNU-injured retina

To examine whether ZDE can protect the degenerated photoreceptors, we stained retinal sections with DAPI and measured the thickness of the outer nuclei layer (ONL) where photoreceptors are located and also counted the row of cells in ONL. Examples of an enlarged region at 1mm away from the optic nerve center of the C-cup of the retinal section from the solvent-treated and 9mg/kg ZDE were shown in Fig 3A. ZDE enhanced the thickness of ONL of MNU-injured retina.

Applying various doses of ZDE showed that with the increased dose of ZDE, the ONL thickness was also increased with a peak at 9 mg/kg of ZDE. The ONL thickness was increased from 7.2 ± 3.2 μm in the solvent group to 15.8 ± 5.4 μm at 1 mg/kg, then 21.2 ± 3.2 μm at 3 mg/kg (p <0.05 vs. solvent), and 29.4 ± 3.3 μm at 9 mg/kg (p<0.001 vs. solvent) then 21.2 ± 1.6 μm at 27 mg/kg. (Fig 3B). We further analyzed the ONL thickness and number of cell layers from center to peripheral for ZDE at 9mg/kg (Fig 3C). ZDE increased the number of cell layers as well as the ONL thickness in most regions along the c-cup, although still worse than those in normal mice. As 9 mg/kg gave the best protection for photoreceptor survival, we chose this dose for further experiments.

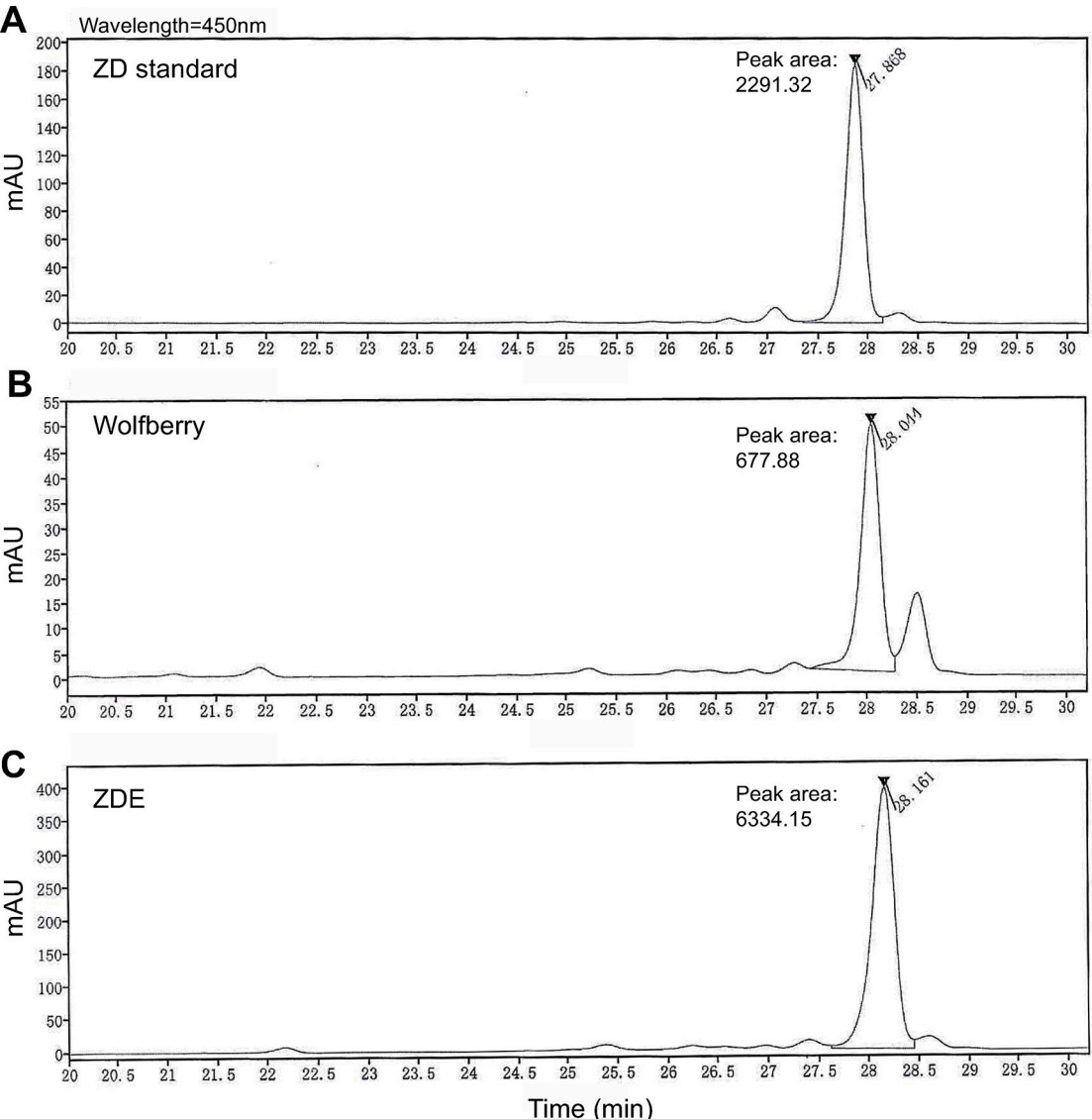

**Fig 2. The ZDE product contains higher ZD than dry wolfberry.** The HPLC chromatograms show the peak of ZD from ZD standard (**A**), dry wolfberry (**B**), and the ZDE product (**C**). The peak time is 27.868, 28.011, and 28.161 mins for ZD standard, wolfberry, and ZDE respectively.

We further tested the safety of the ZDE and the solvent corn oil on normal mice. Both the number of ONL nuclei layers and the ONL thickness were similar among normal mice and those treated with solvent or ZDE at 9mg/kg (S2A Fig).

## ZDE enhances the retinal light responses of MNU-injured mice

To examine whether ZDE can further improve the visual function of MNU-injured mice, we first performed electroretinogram recording (ERG) to examine the retinal light responses. Compared with the normal control mice with large ERG responses, only weak ERG responses remained under both dark adaptation (i.e. scotopic) and light adaptation (i.e. photopic) conditions in the solvent-treated MNU injured group. In contrast, clear a-wave (group responses of photoreceptors) and b-waves (group responses of bipolar cells) could be observed in ZDE-

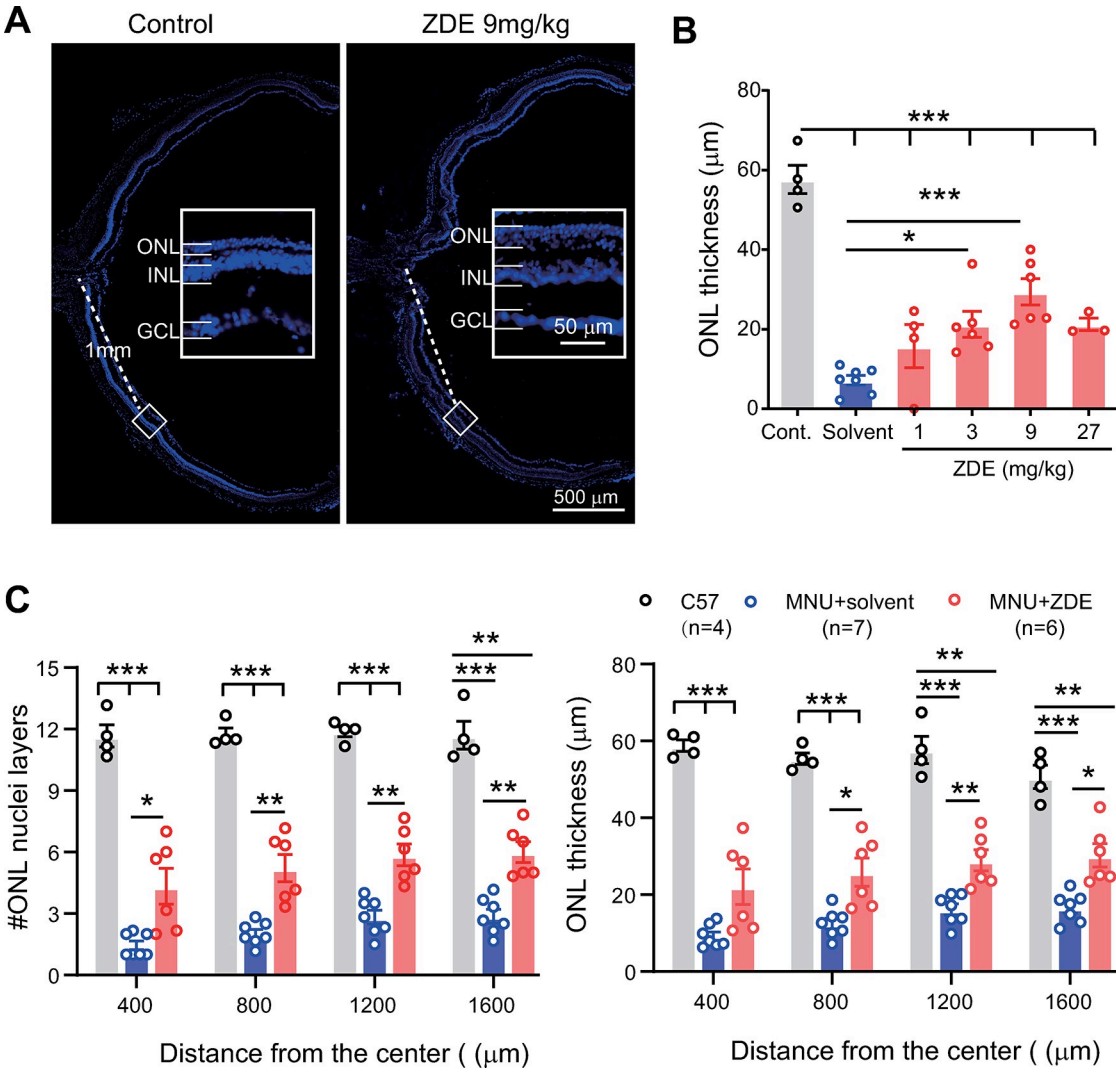

**Fig 3. ZDE improves photoreceptor survival in MNU-injured retina. (A)**. Images of retinal sections in full length as C-cup from solvent treated and ZDE (at 9mg/kg) treated MNU-injured mice. Insets showed the enlarged region at 1mm away from the optic never center. **(B)**. Average thickness of ONL of normal control retina and MNU-injured retina after treatment of solvent or ZDE at increasing doses. The number of animals tested is 4, 7, 4, 6, 6 and 3 for control, solvent and 1, 3, 9, 27 mg/kg ZDE respectively. **(C)**. Average number of ONL cell layers (left) and thickness of ONL(right) of normal control retina and MNU-injured retina treated with solvent or 9 mg/kg ZDE at various distance away from the center of optic nerve. ONL, outer nuclei layer; INL, inner nuclei layer; GCL, ganglion cell layer. *, $p<0.05$, **, $p<0.01$, ***, $p<0.001$, one-way ANOVA followed by Dunnett's multiple comparison test for B and two-way ANOVA test followed by Tukey's multiple comparison test for C.

treated animals (examples shown in Fig 4A). Analyzing the amplitudes of a- and b-waves showed a significant improvement by ZDE at strong flash intensities (Fig 4B and 4C) under scotopic condition, though much weaker than normal control. For example, at scotopic 3.0 cd. s/m$^2$, the amplitude of a-wave was significantly improved from 7.5 ± 1.8 μV at solvent to 43.9 ± 15.7 μV at 9 mg/kg ZDE ($p<0.001$), and the b-wave amplitude was improved from 5.7 ± 1.8 μV at solvent to 90.4 ± 31.2 μV at 9 mg/kg ZDE ($p<0.001$). But ZDE hardly helped the photopic responses. This indicates that ZDE enhanced light responses from the rod pathway but not the cone pathway.

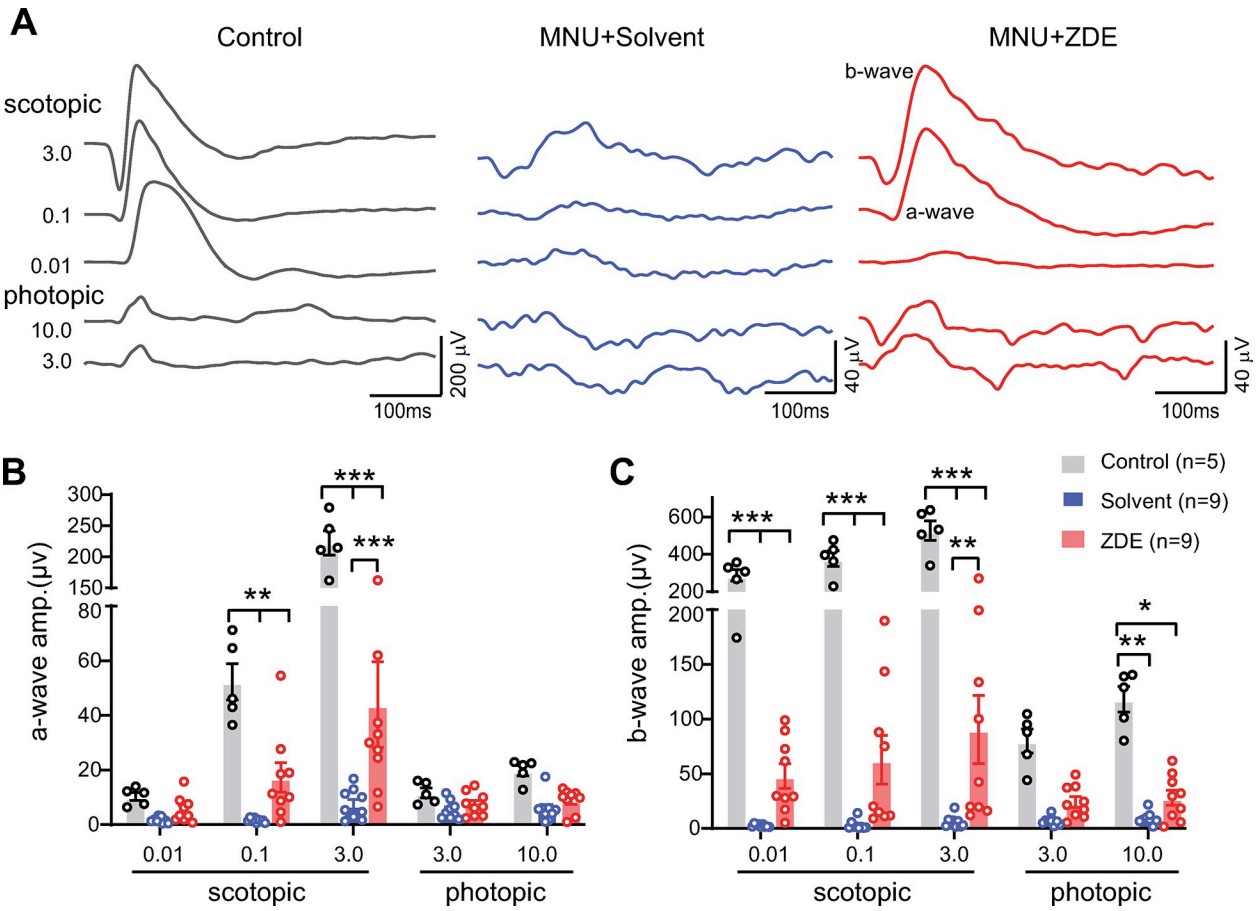

**Fig 4. ZDE enhances the retinal light responses of MNU-injured mice. (A)**. Example traces of ERG recording from normal control, solvent and ZDE-treated MNU-injured mice. Values on the left show the flash intensity with the unit of cd.s/m$^2$ under either scotopic (i.e. dark adaptation) or photopic (i.e. light adaptation) conditions. Peaks of a- and b-waves were shown in ZDE-treated animals. Note the different scales of control and MNU-injured groups. **(B, C)**. Average amplitudes of a-wave **(B)** and b-wave **(C)** for all-flash conditions. *, $p < 0.05$, **, $p < 0.01$; ***, $p < 0.001$, two-way ANOVA test of repetitive measurement followed by Tukey's multiple comparison test.

Also for the normal C57 mice treated with solvent or 9mg/kg ZDE, their retinal light responses were similar to those of untreated normal C57 mice (S2B Fig). This further supported that the solvent and 9mg/kg ZDE application were safe for normal mice retina.

## ZDE enhances the visual acuity of MNU-injured mice

As ZDE protected the degenerated photoreceptors, we then examined whether ZDE can protect the MNU-injured mice against vision loss. We first applied the dark-light-transition box which monitors the ability of animals to tell luminance (Fig 5A). Normal mice tend to spend a longer time in the dark box than the light box (with ~72% time in the dark box). The MNU-injured mice, however, spent ~65% time in the dark box (p = 0.096 vs. WT), and ZDE didn't improve the time in the dark box (Fig 5B).

We further applied the optomotor test to monitor the visual acuity of the animals (Fig 5C). The mouse can track the direction of rotating gratings with the head movement if it can see it (i.e. the optokinetic reflex). The higher the spatial frequency of the grating (i.e. the thinner the grating) that could trigger the optokinetic reflex of the mouse, the higher the visual acuity of the animal [21]. Normal mice have a visual acuity of 0.4 cycles/degree (cpd), MNU injured mice

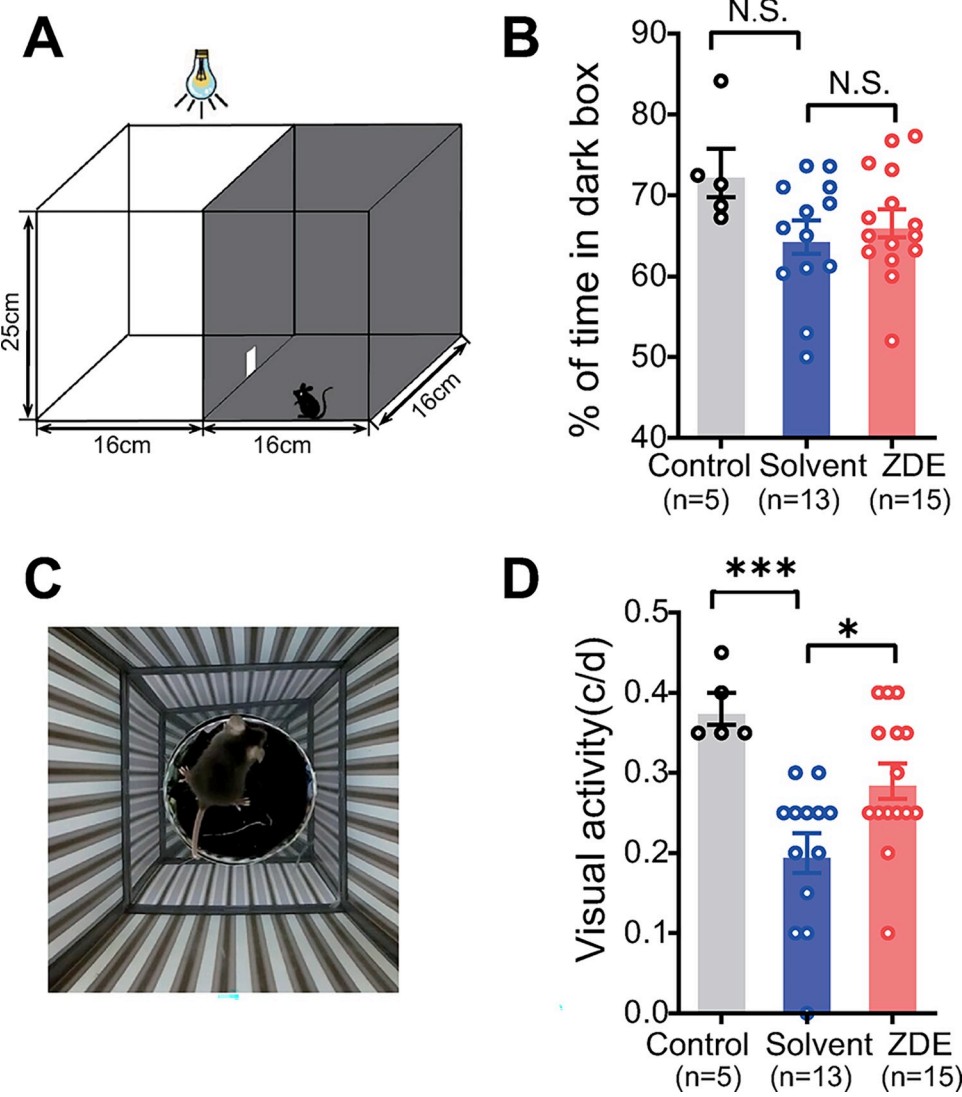

**Fig 5. ZDE enhances the visual acuity of MNU-injured mice. (A).** Illustration of the dark-light-transition box. **(B).** Average percent of time spent in the dark chamber. **(C).** Illustration of the optomotor system. **(D).** Average visual acuity of normal mice, solvent and ZDE-treated MNU-injured mice. *, p<0.05, ***,p<0.001, one-way ANOVA test followed by Tukey's multiple comparison test. N.S., no significant difference.

had impaired visual acuity around 0.2 cpd, which was significantly improved by ZDE treatment (Fig 5D). Therefore, ZDE improved the impaired visual functions of MNU-injured mice.

## Discussion

In the current study, we developed a method to enrich ZD from the dry fruit of wolfberry and then used MNU-injured mice as a model of RP and treated them with the enriched product. Our results showed that ZDE effectively improved the survival of photoreceptors and visual function of injured mice. This is a continuous work of our previous study, where one shot of ZD (95% purity) injection intravitreally rescued the degenerated retina of rd10 mice [15]. Our study proved that similar to the purified ZD, our ZDE product also demonstrated a neuroprotective role in photoreceptor degenerated retina.

Wolfberry extract has been shown to protect retinal neurons in various retinal diseases in both animal studies [8–10] and clinical studies [12]. Its major antioxidant components include water soluble component Lycium barbarum polysaccharides (LBP), and water insoluble components flavonoids and carotenoids. A recent clinical study that used wolfberry extract (with a main active ingredient as LBP) to treat RP patients showed that both the ERG and the visual acuity of these patients have been remarkably improved [12]. Lycium barbarum glycopeptide (LbGP), immunoreactive glycoproteins extracted from LBP have also been shown to protect MNU-injured photoreceptors in mouse models [22]. The product of LbGP is on the market now as a supplement to improve vision.

Another group of major constituents in wolfberry are carotenoids, which include zeaxanthin and its di-esters, ZD. ZD is much more abundant in wolfberries than zeaxanthin [23]. It accounts for 31%–56% of carotenoids in wolfberry and 0.01–0.2% of the mature fruit of wolfberry [24]. With multiple conjugated double bonds in its chemical structure, ZD has a strong anti-oxidant function as demonstrated by its strong ability to scavenge free radicals and protect liver cells against several liver diseases [13,14]. The retinal protective effect of zeaxanthin (together with lutein) has been shown in many animal and clinical studies [25–27]. Our recent work further showed that intravitreous injection of pure ZD can effectively protect degenerated photoreceptors [15]. It not only improved the visual behavior of rd10 mice (a genetic mutation mouse model of RP) but also improved the light responses of photoreceptors, bipolar cells, and retinal ganglion cells. ZD also reduced the upregulated expression of genes that are involved in inflammation, apoptosis, and oxidative stress in the rd10 retina. ZD further reduced the activation of two key factors, STAT3 and CCL2, down-regulated the expression of the inflammatory factor GFAP, and inhibited ERK, and P38 pathways [15]. This study suggests that ZD might be the other most important component of wolfberry in RP treatment besides LbGP.

However, in the previous study, only one shot of intravitreous injection was applied to rd10 mice, a better protective effect would be expected if the animal could be treated with more frequent feedings. However, due to the long and expensive procedure of purifying ZD from wolfberry, the supply of pure ZD was limited. This was why we started this study by working with the industry to extract large amounts of ZD from wolfberry to enable enough supply of ZDE for oral administration. It's worth noting that the metabolic pathway is different between oral feeding and intravitreous injection, so the protective effect of ZDE may act through different active pathways than intravitreous injection.

In this study, ZDE produced from the production line contained ~3% ZD in the extract, which is over 9 times more concentrated than the raw fruit. The protective dose of ZDE at 9mg/kg (of ZD) was within the range of other studies which showed that 2mg -10mg/kg oral feeding of pure ZD effectively alleviated hepatic injury [13,28]. It is worth noting that besides ZD in the ZDE, there might be other beneficial components of wolfberry that help to protect retinal neurons, since the ZDE product contains Gouqi oil including mainly fatty acid like linolenic acid and linoleic acid, and a few others like palmitic acid, stearic acid and oleic acid. Therefore, ZDE may have advantages over the purified ZD to treat RP as a supplement, not only because of the much cheaper cost and production time but also because of the mixture of other potential beneficial components.

The current research has several limitations to consider. First, we only evaluated the survival of photoreceptors by DAPI staining, the structure of photoreceptors can be better disclosed by immunostaining for the outer segment of rods or cone with rhodopsin or opsin. And H&E staining can also help to disclose the detailed structure of plexiform layer of the retina besides the soma. Second, the protective effect of ZDE on the visual behavior was not as significant as ERG responses. ZDE only slightly (though significantly) improved the visual

acuity of the mice, but failed to improve the ability to tell luminance. This may be due to the limited effect of ZDE on the destructed inner retinal circuit by MNU (which processes visual information that lead to the visual behavioral responses), even though it enhanced the impaired outer retinal structure (which contributes to the ERG responses). Immunostaining of the inner retinal structure may be further carried out to explore the reason. Third, we only examined the effect of ZDE on MNU-induced photoreceptor degeneration for 14 days, and its protective effect in the long term is unknown. Also while we showed that daily oral given ZDE does not influence retinal functions, it's a short-term toxicity test. It's not clear whether there would be long-term toxicity or side effects since in the clinic RP is a chronic progression and the patient may need long-term treatment. Fourth, we pre-treated the animal with ZDE before the MNU injury, to get a good protective effect. However, in the clinic, treatment can only be applied after the diagnosis of the disease. Therefore, it's important to test whether the application of ZDE after MNU injury may also delay photoreceptor degeneration. Fifth, it is important to examine the pharmacokinetics of ZDE after oral feeding, since the large molecular weight of ZD (over 1000) may affect its ability to penetrate the blood-retina-barrier. Indeed, we have tried to collect the retina and vitreous of the mice at various time points after feeding them with ZDE but failed to get any good HPLC results from these samples. Experience in extracting ZD from the tissues is needed for the test, and a more sensitive method such as mass spectrum may be applied to detect the low concentration of ZD in the eye if it enters. We're still working on this.

## Conclusion

We developed a way to enrich ZD from wolfberry in large amounts, and this ZDE product can protect injured retinal photoreceptors in mouse model.

## Supporting information

**S1 Fig. The body weight of the animal remained stable during the treatment.** The record of mice's body weight over 2 weeks of ZDE treatment at various doses.
(TIF)

**S2 Fig. No impact of solvent or ZDE on the normal retina. A.** Average number of ONL nuclei layers (left) and thickness of ONL (right) of normal C57 retina and those treated with solvent or 9 mg/kg ZDE. **B.** Average amplitudes of a-wave (left) and b-wave (right) for normal C57 mice and those treated with solvent or 9mg/kg ZDE.
(TIF)

## Author Contributions

**Conceptualization:** Wenhua Zhang, Jinhong Zhang, Xiangfeng Hao, Kwok-Fai So, Ying Xu.

**Data curation:** Xiongmin Chen, Sensen Zhang, Lili Yang, Qihang Kong.

**Formal analysis:** Xiongmin Chen, Sensen Zhang, Lili Yang, Qihang Kong.

**Funding acquisition:** Wenhua Zhang, Ying Xu.

**Investigation:** Xiongmin Chen, Sensen Zhang, Qihang Kong, Ying Xu.

**Methodology:** Xiongmin Chen, Sensen Zhang, Qihang Kong, Kwok-Fai So, Ying Xu.

**Project administration:** Kwok-Fai So, Ying Xu.

**Resources:** Lili Yang, Wenhua Zhang, Jinhong Zhang, Xiangfeng Hao, Kwok-Fai So, Ying Xu.

**Supervision:** Jinhong Zhang, Xiangfeng Hao, Kwok-Fai So, Ying Xu.

**Validation:** Xiongmin Chen, Kwok-Fai So.

**Writing – original draft:** Ying Xu.

**Writing – review & editing:** Kwok-Fai So, Ying Xu.

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
