## [Decision Letter · Decision Letter 0]

20 Feb 2024

PONE-D-24-04375Zeaxanthin dipalmitate-enriched wolfberry extract improves vision in a mouse model of photoreceptor degenerationPLOS ONE

Dear Dr. Xu,

Thank you for submitting your manuscript to PLOS ONE. After careful consideration, we feel that it has merit but does not fully meet PLOS ONE’s publication criteria as it currently stands. Therefore, we invite you to submit a revised version of the manuscript that addresses the points raised during the review process.

We look forward to receiving your revised manuscript.

Kind regards,

Kin-Sang Cho, PhD

Academic Editor

PLOS ONE

Journal Requirements:

2. Please provide additional information about the source of the wolfberry used in this study. If plant material was collected, please include data such as the geographic location, details of any permits, information about voucher specimens and any authentication methods. If material was purchased from a company or obtained from a third party, we would expect information such as the name and location of the company/party supplying the product, a product description and name, product numbers, lot numbers, quality assessments etc.

3. To comply with PLOS ONE submissions requirements, in your Methods section, please provide additional information regarding the experiments involving animals and ensure you have included details on (1) methods of sacrifice, (2) methods of anesthesia and/or analgesia, and (3) efforts to alleviate suffering.

4. We note that you have a patent relating to material pertinent to this article. Please provide an amended statement of Competing Interests to declare this patent (with details including name and number), along with any other relevant declarations relating to employment, consultancy, patents, products in development or modified products etc. Please confirm that this does not alter your adherence to all PLOS ONE policies on sharing data and materials, as detailed online in our guide for authors http://journals.plos.org/plosone/s/competing-interests by including the following statement: "This does not alter our adherence to  PLOS ONE policies on sharing data and materials.” If there are restrictions on sharing of data and/or materials, please state these. Please note that we cannot proceed with consideration of your article until this information has been declared.

“This work was funded by Ningxia Key Research and Development Program Grant (No. 2021BEF02040 to WZ) and the Natural Science Foundation of Guangdong Province (2023A1515012397 to YX).”

“This work was funded by Ningxia Key Research and Development Program Grant (No. 2021BEF02040 to WZ) and the Natural Science Foundation of Guangdong Province (2023A1515012397 to YX).”

6. In the online submission form, you indicated that [Insert text from online submission form here].

7. Your ethics statement should only appear in the Methods section of your manuscript. If your ethics statement is written in any section besides the Methods, please delete it from any other section.

Reviewers' comments:

Reviewer's Responses to Questions

**Comments to the Author**

1. Is the manuscript technically sound, and do the data support the conclusions?

Reviewer #1: Yes

Reviewer #2: Yes

Reviewer #3: Yes

2. Has the statistical analysis been performed appropriately and rigorously? 

Reviewer #1: Yes

Reviewer #2: Yes

Reviewer #3: Yes

3. Have the authors made all data underlying the findings in their manuscript fully available?

Reviewer #1: Yes

Reviewer #2: Yes

Reviewer #3: Yes

4. Is the manuscript presented in an intelligible fashion and written in standard English?

Reviewer #1: Yes

Reviewer #2: Yes

Reviewer #3: Yes

5. Review Comments to the Author

Reviewer #1: The goal of the authors of the manuscript is to describe a new way to produce extract from Wolfberry containing higher amount of Zeaxanthin dipalmitate (ZD) and to test the effects of this extract (ZDE) in an in vivo model of retinal degeneration induced by intraperitoneaal injection of N-methyl-N-nitrosourea in C57BL6 mice.

first the =y evaluate the quantity and purity of ZD in their extract produced by this new technique and then they evaluate the effect of oraal administration fro 14 days of ZDE on retinal structure, photoreceptor survival, ERG response of the retina submited to several type of stimuli (at various frequencies and/or intensities) and then the evaluate the visulaa function in the whole animal using two different tests in addition they provide certain proofs of context showing the sfatey of the doses of the ZDE used in the supplemental figures.

in general although simple the paper is clear, the methods used classic but pertinent to answer the questions raised and the data tends to support the claims.

the methods used and the statistical analysis seems accurate as well.

i have only minor comments to improve the manuscript:

1 figure Suplementary S2 the file is Corrrupted and cannot be read appropriately and need to be replaced.

the abstract and the intro needs to be checked for english as some sentence may be awkward:

line 35 this sentence needs to rephrased :The adult C57BL/6 mice were treated with ZDE or solvent by daily gavage for 2 weeks, during the middle time (?) the animals were intraperitoneally injected with N-methyl-N-nitrosourea to induce photoreceptor degeneration.

intro line 58: While these products ... needs to be replaced by while this strategy using gene therapy...

In the methods line 106 to 108 the sentence needs to be reformulated for more clarity

line 141 give the figures for the 9mg/kg body weight optimal treatment concentration

in the discussion line 326 : add imporved the survival (of photoreceptors)

in the discussion they say that LgBP is recognized as the main antioxidant molecule in wolfberry. What is the concentration of LbGP in their ZDE ? the authors needs to discuss at the least the others compound present or potentially present in their extract. any HPLC data on this?

i wanted them to discuss why they have no effect on their dark and light bi=ox test evaluating the ability of mice to see difference of luminance while their experiments show an effect on scotopic vision.

Apparently this test is not sensitive enough and does not allow to see a difference between normal mice an mice with retinal degenration. therefore it must be discussed and if it is the case this test should be discarded from the article.

Reviewer #2: Chen et al examined the impact of zeaxanthin dipalmitate (ZD)-enriched wolfberry extract on photoreceptor health in mice with N-methyl-N-nitrosourea-induced retinal degeneration. This is a continuous study from their previous work which has shown one shot of intravitreal ZD (95% purity) rescued photoreceptors in rd10 mice. Overall, the current work is well-designed and the manuscript is well-written. There are only several minor comments to address.

1. Fig 5B: statistics analysis shall be shown and “not significant” shall be labeled on the graph if no statistical significance was found. The statement at line 306-308 shall be toned down if there was no significance.

2. Supplement Figure 2 was too small to see. The authors please correct. The graph shall be exported in JPEG or Tiff. Same as S. Figure 1.

3. Figure legends were embedded in the main text. It shall be moved after references.

4. Line 336, please create the reference including authors, title, journal and publication date for the article under press.

Reviewer #3: Chen et. al. used the their new developed method enriched ZD from wolfberry and test the efficacy of ZDE in MNU-injured retina. The enriched ZDE showed 9 times higher conc. than that in wolfberry. And authors conducted dose response to optimize the dose selection. By daily pretreating mice before MNU injury and until the end of study with ZDE, they demonstrated the neuroprotective effects on MNU-injured retina. It is well-designed study. Below please find my TWO additional comments, to make it better.

1. In the referred previous study by authors' group on ZD, they performed intravitreal injection but not oral treatment. It is understandable that oral given will need high dose, but the metabolic pathway is different. The active pathway thereby could be different with previous study. Can authors add one paragraph in the discussion part?Also, did authors test if ZD reached to the retina post oral treatment? And how much? The molecular weight of ZD is over 1000. Would it affect ZD penetrate the BRB if give orally?

2. How about the purity of ZDE? There are some small peaks in Fig.2C.

3. In supplementary data, authors presented that oral given ZDE has no influence on retinal functions. But it is a short-term toxicity test. In clinic, RP has chronic progression, which is different with RD10 or MNU-injured model. It may need long-term drug given. Have the authors tested the long-term toxicity or side effects by daily oral given?

6. PLOS authors have the option to publish the peer review history of their article (what does this mean?). If published, this will include your full peer review and any attached files.

Reviewer #1: No

Reviewer #2: No

Reviewer #3: No

---

## [Author Response · Author response to Decision Letter 0]

3 Apr 2024

Reviewer #1: The goal of the authors of the manuscript is to describe a new way to produce extract from Wolfberry containing higher amount of Zeaxanthin dipalmitate (ZD) and to test the effects of this extract (ZDE) in an in vivo model of retinal degeneration induced by intraperitoneaal injection of N-methyl-N-nitrosourea in C57BL6 mice.

first the =y evaluate the quantity and purity of ZD in their extract produced by this new technique and then they evaluate the effect of oraal administration fro 14 days of ZDE on retinal structure, photoreceptor survival, ERG response of the retina submited to several type of stimuli (at various frequencies and/or intensities) and then the evaluate the visulaa function in the whole animal using two different tests in addition they provide certain proofs of context showing the sfatey of the doses of the ZDE used in the supplemental figures.

in general although simple the paper is clear, the methods used classic but pertinent to answer the questions raised and the data tends to support the claims.

the methods used and the statistical analysis seems accurate as well.

i have only minor comments to improve the manuscript:

1 figure Suplementary S2 the file is Corrrupted and cannot be read appropriately and need to be replaced.

[A] Sorry, the Supplementary S2 is now replaced.

the abstract and the intro needs to be checked for english as some sentence may be awkward:

line 35 this sentence needs to rephrased :The adult C57BL/6 mice were treated with ZDE or solvent by daily gavage for 2 weeks, during the middle time (?) the animals were intraperitoneally injected with N-methyl-N-nitrosourea to induce photoreceptor degeneration.

[A] It’s now clarified. 

intro line 58: While these products ... needs to be replaced by while this strategy using gene therapy...

[A] It’s corrected as suggested.

In the methods line 106 to 108 the sentence needs to be reformulated for more clarity

[A] It's now clarified.

line 141 give the figures for the 9mg/kg body weight optimal treatment concentration

[A]Sorry that we can’t provide the HPLC result for the 9mg/kg weight concentration. We only have data of the final ZDE product (which showed ~3mg/g ZD in the product), and the dose of 9mg/kg body weight is made by diluting the ZDE product.

in the discussion line 326 : add imporved the survival (of photoreceptors)

[A] It’s corrected now.

in the discussion they say that LgBP is recognized as the main antioxidant molecule in wolfberry. What is the concentration of LbGP in their ZDE ? the authors needs to discuss at the least the others compound present or potentially present in their extract. any HPLC data on this?

[A] Sorry for the confusion. The LBP is the water soluble components in wolfberry while ZD is oil soluble. When extracting ZD, the polysaccharide extract (i.e. LBP) was in the supernatant and discarded while the precipitate was collected for further processing to get ZD. So there is no LBP in ZDE product. The other components in the ZDE extract is Gouqi oil, which are mainly fatty acid like Linolenic acid and Linoleic acid, and others like palmitic acid, Stearic acid and Oleic acid. I have now clarified this in the Discussion. Sorry there is no HPLC data on this right now, and it will take long time to identify each component by HPLC.

i wanted them to discuss why they have no effect on their dark and light box test evaluating the ability of mice to see difference of luminance while their experiments show an effect on scotopic vision.

[A] Actually it’s discussed on line 382 to 387. We think the discrepancy between luminance detection and scotopic ERG may be due to the limited effect of ZDE on the destructed inner retinal circuit even though it enhanced the impaired outer retinal structure (which produced a- and b-waves in ERG) . 

Apparently this test is not sensitive enough and does not allow to see a difference between normal mice an mice with retinal degeneration. therefore it must be discussed and if it is the case this test should be discarded from the article.

[A]The dark and light box test mainly tell how well the animal can respond to luminance. Although it may not be as sensitive as optomotor test, the retinal degenerated mice indeed performed worse than the normal mice in this test, as shown in various models, including rd10 mice (Zhang J, Exp Eye Res, 2017; Xiang Z, Neuropharmacology, 2018; Liu XB, Neural Regeneration Reserach, 2021), rd1 mice (Wang Y et al., Cellular Physiolgy and Biochemsitry, 2017; Liu F et al., IOVS, 2018 ), NaIO3 injured mice (Espitia-Arias et al., Antioxidtants, 2023) and so on. In our current study, MNU-injured model was applied. The MNU-injured mice tent to spend less time (64.8 ± 7.4%) than WT (72.8 ± 6.7%) in black area (p=0.096). The reason of reaching significant difference may be due to the limited number of WT mice tested, and indeed in our newly published work by Kong Q et al ( Neural Regeneration Research, 2024), the difference between MNU injured mice and WT mice was obvious. So we think it’s important to keep this test as one indicator for the visual behavior. 

Reviewer #2: Chen et al examined the impact of zeaxanthin dipalmitate (ZD)-enriched wolfberry extract on photoreceptor health in mice with N-methyl-N-nitrosourea-induced retinal degeneration. This is a continuous study from their previous work which has shown one shot of intravitreal ZD (95% purity) rescued photoreceptors in rd10 mice. Overall, the current work is well-designed and the manuscript is well-written. There are only several minor comments to address.

1. Fig 5B: statistics analysis shall be shown and “not significant” shall be labeled on the graph if no statistical significance was found. The statement at line 306-308 shall be toned down if there was no significance.

[A] So all the graphs are labelled with “NS”, and in the text I added the p value to indicate the statistical difference.

2. Supplement Figure 2 was too small to see. The authors please correct. The graph shall be exported in JPEG or Tiff. Same as S. Figure 1.

[A] Sorry it’s a mistake. It’s now replaced.

3. Figure legends were embedded in the main text. It shall be moved after references.

[A] According to the style request of Plos ONE, “Each figure caption should appear directly after the paragraph in which they are first cited”, so I have to keep them in the main text. 

4. Line 336, please create the reference including authors, title, journal and publication date for the article under press.

[A] It’s now published so the reference is added.

Reviewer #3: Chen et. al. used the their new developed method enriched ZD from wolfberry and test the efficacy of ZDE in MNU-injured retina. The enriched ZDE showed 9 times higher conc. than that in wolfberry. And authors conducted dose response to optimize the dose selection. By daily pretreating mice before MNU injury and until the end of study with ZDE, they demonstrated the neuroprotective effects on MNU-injured retina. It is well-designed study. Below please find my TWO additional comments, to make it better.

1. In the referred previous study by authors' group on ZD, they performed intravitreal injection but not oral treatment. It is understandable that oral given will need high dose, but the metabolic pathway is different. The active pathway thereby could be different with previous study. Can authors add one paragraph in the discussion part?

[A] Agree, but the active pathway is not clear so far to explain well, so we only added a sentence in the Discussion to address this fact.

Also, did authors test if ZD reached to the retina post oral treatment? And how much? The molecular weight of ZD is over 1000. Would it affect ZD penetrate the BRB if give orally?

[A] The large molecular weight of ZD may affect its penetration of blood-retina-barrier. Indeed we have tried to examine the pharmacokinetics of ZDE after oral feeding the animal, but failed to get any good HPLC results from the collected samples of vitreous and retinas. A more sensitive method such as mass spectrum may be applied to detect the low concentration of ZD in the eye if it enters. So we can’t answer the question right now. We have addressed this limitation in the Discussion.

2. How about the purity of ZDE? There are some small peaks in Fig.2C.

[A] I’m not sure whether I understood the question correctly. The small peaks of ZD from ZDE product presented at the same locations as the ZD standard with a purity of ~95%（Fig.2A). So we think the purity of ZDE is similar as the standard, which is ~95%. On the other hand, the ZDE product we made contains ~3% of ZD, with the rest component as Gouqi oil, which I included mainly fatty acid like linolenic acid and linoleic acid, and a few others like palmitic acid, stearic acid and oleic acid.

3. In supplementary data, authors presented that oral given ZDE has no influence on retinal functions. But it is a short-term toxicity test. In clinic, RP has chronic progression, which is different with RD10 or MNU-injured model. It may need long-term drug given. Have the authors tested the long-term toxicity or side effects by daily oral given?

[A] Sorry, no, we didn’t test the long-term toxicity of daily oral given ZDE. We have now added a sentence to address this limitation in the Discussion.

---

## [Decision Letter · Decision Letter 1]

11 Apr 2024

Zeaxanthin dipalmitate-enriched wolfberry extract improves vision in a mouse model of photoreceptor degeneration

PONE-D-24-04375R1

Dear Dr. Xu,

We’re pleased to inform you that your manuscript has been judged scientifically suitable for publication and will be formally accepted for publication once it meets all outstanding technical requirements.

Kind regards,

Kin-Sang Cho, PhD

Academic Editor

PLOS ONE

Additional Editor Comments (optional):

Reviewers' comments:

Reviewer's Responses to Questions

**Comments to the Author**

1. If the authors have adequately addressed your comments raised in a previous round of review and you feel that this manuscript is now acceptable for publication, you may indicate that here to bypass the “Comments to the Author” section, enter your conflict of interest statement in the “Confidential to Editor” section, and submit your "Accept" recommendation.

Reviewer #1: All comments have been addressed

Reviewer #2: All comments have been addressed

Reviewer #3: All comments have been addressed

2. Is the manuscript technically sound, and do the data support the conclusions?

Reviewer #1: Yes

Reviewer #2: Yes

Reviewer #3: Yes

3. Has the statistical analysis been performed appropriately and rigorously? 

Reviewer #1: Yes

Reviewer #2: Yes

Reviewer #3: Yes

4. Have the authors made all data underlying the findings in their manuscript fully available?

Reviewer #1: Yes

Reviewer #2: Yes

Reviewer #3: Yes

5. Is the manuscript presented in an intelligible fashion and written in standard English?

Reviewer #1: Yes

Reviewer #2: Yes

Reviewer #3: Yes

6. Review Comments to the Author

Reviewer #1: No specific comments the authors responded satifyingly to the comments of all three reviewers comments.

Reviewer #2: The authors have addressed my concerns in the current revised manuscript. I have no additional comments.

Reviewer #3: Authors have addressed my concerns and discussed in the discussion part. I would suggest to accept.

7. PLOS authors have the option to publish the peer review history of their article (what does this mean?). If published, this will include your full peer review and any attached files.

Reviewer #1: No

Reviewer #2: No

Reviewer #3: No

---

## [Editor Report · Acceptance letter]

8 May 2024

PONE-D-24-04375R1 

PLOS ONE

Dear Dr. Xu, 

I'm pleased to inform you that your manuscript has been deemed suitable for publication in PLOS ONE. Congratulations! Your manuscript is now being handed over to our production team.

Kind regards, 

on behalf of

Kin-Sang Cho 

Academic Editor

PLOS ONE